# Controllable two-dimensional movement and redistribution of lithium ions in metal oxides

Xiufeng Tang[1,2], Guoxin Chen[1,2], Zhaopeng Mo[1,2], Dingbang Ma[1,2], Siyuan Wang[1,2], Jinxiu Wen[1,2], Li Gong[3], Lite Zhao[1], Jingcheng Huang[1,2], Tengcheng Huang[1,2] & Jianyi Luo[1,2]

Rechargeable lithium batteries are the most practical and widely used power sources for portable and mobile devices in modern society. Manipulation of the electronic and ionic charge transport and accumulation in solid materials has always been crucial for rechargeable lithium batteries. The transport and accumulation of lithium ions in electrode materials, which is a diffusion process, is determined by the concentration distribution of lithium ions and the intrinsic structure of the electrode material and thus far has not been manipulated by an external force. Here, we report the realization of controllable two-dimensional movement and redistribution of lithium ions in metal oxides. This achievement is one kind of centimeter-scale control and is achieved by a magnetic field based on the 'current-driving model'. This work provides additional insight for building safe and high-capacity rechargeable lithium batteries.

[1] School of Applied Physics and Materials, Wuyi University, Jiangmen 529020, China. [2] Research Center of Flexible Sensing Materials and Devices, Wuyi University, Jiangmen 529020, China. [3] Instrumental Analysis & Research Center, Sun Yat-sen University, Guangzhou 510275, China. Correspondence and requests for materials should be addressed to J.L. (email: luojiany@mail3.sysu.edu.cn)

In the field of electrochemistry, the development of intercalation theory indicates that fast transport and accumulation of lithium (Li) ions in solid materials can be achieved[1]. This discovery directly gave rise to secondary (rechargeable) lithium batteries, such as Li-metal batteries[2,3], Li-ion batteries[4,5], and Li-polymer batteries[6,7]. Tremendous efforts and great progress have been made in building better batteries by exploring better intercalation compounds[4,5,8–10], monitoring the solid-electrolyte interface during the intercalation process[11–13], modifying the interface with a solid-electrolyte interphase (SEI)[14–16], and improving the electrical conductivity of anodes to enhance electron transport[17–19]. Based on the large gravimetric and volumetric energy densities of rechargeable lithium batteries, they have become the most practical and widely used power sources for portable and mobile applications[20–22].

Notably, manipulation of the transport and accumulation of Li ions in electrode materials has always been crucial for rechargeable lithium batteries[23,24]. The ionic and electronic transport processes determine the maximum power output and the minimum charging time of a lithium-ion battery, and ionic diffusion within the electrodes generally represents a fundamental limitation of the rate at which a battery can be charged and discharged[5]. Accordingly, intensive research has been directed toward monitoring the diffusion process and mapping the spatial distribution of Li ions in electrode materials to provide a fundamental understanding of the transport and accumulation process[25–29]. In addition, further attempts to manipulate the distribution of Li ions in electrode materials are underway. Recently, the concept of an ion redistributor was proposed to regulate the movement of Li ions to deliver a uniform Li-ion distribution for dendrite-free Li deposition[30]. The stabilization of superdense lithium storage between two graphene sheets was surprisingly achieved as its appearance is typically reserved[31]. However, the planar movement and distribution of Li ions in electrode materials still cannot be controlled by an external force. For this reason, rechargeable lithium batteries face numerous problems and safety concerns, with Li dendrites as the most representative example[32,33].

Here, we report the controllable two-dimensional (2D) movement and redistribution of Li ions in metal oxides by a magnetic field based on the 'current-driving model'. There are three prerequisites in the 'current-driving model': a solid–liquid interface with dynamic exchange of Li ions, a magnetic field, and work being done on the system. This ubiquitous 'current-driving model' can be used to control Li ions in different metal oxides ($WO_3$, $TiO_2$, $Nb_2O_5$, and $MoO_3$) and also has been proven work for controlling other cations ($H^+$, $Na^+$, $Zn^{2+}$, and $Ca^{2+}$).

## Results

**Formation of the 'current-driving model'**. Both the self-discharge phenomenon in batteries[34–40] and the self-bleaching behavior in electrochromic devices[41,42] are ubiquitous. Thus, we made a reasonable assumption regarding the solid (S)-liquid (L) interface between a Li-intercalated metal oxide and a liquid electrolyte; i.e., the surface barrier of the liquid side ($E_L$) is high, while the counterpart for the solid side ($E_S$) is lower than $E_L$ (Fig. 1, the amplified part). Hence, Li ions in the electrolyte must overcome the high barrier $E_L$ to intercalate into the metal oxide and the low barrier $E_S$ prevents the intercalated Li ions from escaping out of the metal oxide. However, some "outstanding" Li ions (with an initial velocity $V_0$) exist among the intercalated ones, possessing sufficient kinetic energy to overcome the low barrier $E_S$, and spontaneously escape into the electrolyte. Once they reach the electrolyte side, the Li ions that spontaneously escaped will be accelerated by the barrier difference ($\Delta E$) between $E_L$ and $E_S$ and move at a higher velocity ($V$'). Similarly, Li ions

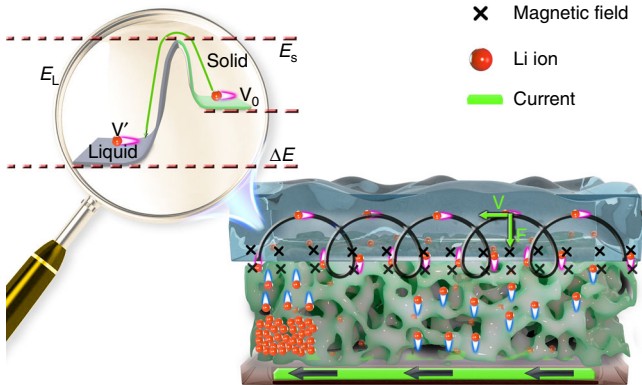

**Fig. 1** Schematic of the scheme to achieve controllable 2D movement of Li ions in a metal oxide. This design is based on a solid–liquid interface between a Li-intercalated metal oxide and a liquid electrolyte and a magnetic field generated by a constant current, which is referred to as the 'current-driving model'. The direction of the current-induced magnetic field at the solid–liquid interface is denoted with symbols in black "X" for the into-paper-plane. The amplified part is a diagram of the assumption about the surface barriers at the solid–liquid interface

from the electrolyte intercalating into the metal oxide will be decelerated by $\Delta E$.

Based on this assumption, Fig. 1 shows a schematic of the scheme to achieve controllable 2D movement of Li ions in a metal oxide. First, Li ions are injected into the metal oxide. Then, the entire system (containing a bottom conducting layer, a Li-intercalated metal oxide, a liquid electrolyte and a solid–liquid interface) is placed in a magnetic field generated by a constant current flowing through the bottom conducting layer. At the solid–liquid interface, the magnetic field is perpendicular to the paper surface facing inward. At this moment, the Li ions at the interface that just escaped from the metal oxide into the electrolyte with velocity $V$', are influenced by a Lorentz force ($\mathbf{F} = q\mathbf{v} \times \mathbf{B}$) pointing to the left and move counterclockwise along a circle with a radius $R$'($R = mv/qB$). Once they complete a half circle and reach the interface again, they ideally will overcome the barrier $E_L$, decelerate to a lower velocity of $V_0$, return to the metal oxide and move counterclockwise along a circle with a smaller radius $R_0$. When a cycle is finished, the Li ions have moved one step toward the left end of the metal oxide due to the difference between the radii ($R$ and $R$'). Once the Li ions at the interface have moved away, the Li ions in the body of the metal oxide will diffuse upward and continuously join the cycle. Thus, under the manipulation of the magnetic field, Li ions will move stepwise toward the left end of the metal oxide and ultimately accumulate there, and their counterparts, the electrons, will be attracted by the Li ions and migrate to the left end of the metal oxide.

In addition, when the distribution of Li ions in the metal oxide changes from a uniformly distributed state to a highly concentrated state, according to the second law of thermodynamics, work must be done on the system because the entropy of the entire system decreases. Therefore, a magnetic field generated by a constant current is preferable, and the current conducting layer is arranged at the bottom of the metal oxide in the system for the convenience of doing work, which is referred to as 'the current-driving model'. Notably, the experimental setup of this model is the same layout as the collector and the electrode in a battery system.

**Delineation of the controllable 2D movement and redistribution of Li ions in metal oxides**. According to the model, experiments were carried out (see "Methods"). To date, controllable 2D

movement and redistribution of Li ions have been achieved in four types of metal oxide nanostructures (both amorphous and crystalline), namely, $WO_3$ films, $MoO_3$ films, $TiO_2$ nanorod films and $Nb_2O_5$ @ graphite nanocomposites (see Supplementary Table 1). Owing to the visualization of Li ions moving in $WO_3$ films (switching between a blue state and a transparent state) i.e., the electrochromic property of $WO_3$ films[43], 2D migration and redistribution of Li ions in an amorphous $WO_3$ film (for film characterization see Supplementary Fig. 1) was used as an example and delineated in detail to elucidate the manipulation of the planar movement and redistribution of Li ions in metal oxides (see Fig. 2). Here, two points should be noted: (i) both the electrochromic property of the $WO_3$ films[44] and the PC $LiClO_4$ electrolyte[45] were used as the only tools to interpret the detailed experimental setup and vividly display the moving process and redistribution of Li ions in metal oxides; (ii) in traditional electrochromic smart windows[46], Li ions in the electrolyte are vertically driven by an electric field that is applied between the upper and bottom transparent conducting layers, and intercalated into the electrochromic film. Then, the whole electrochromic film becomes colored. However, this work begins after Li ions are injected into metal oxides and reports how to control the 2D planar movement and redistribution of Li ions in metal oxides by a magnetic field. Therefore, this work is distinct from traditional electrochromic smart windows.

A sandwiched structure was adopted: upper ITO conducting layer/electrolyte/$WO_3$ film/bottom ITO conducting layer. In P1, a 3 V voltage was applied between the upper and bottom ITO layers; Li ions were vertically injected into the $WO_3$ film, and then the film turned blue (Z-moved state). In P2, a horizontal voltage was applied between the two ends of the bottom ITO layer and a constant current flowed through the bottom ITO layer. In the first cycle, the current was flowing to the right and a magnetic field was generated; at the solid–liquid interface, this field was perpendicular to the paper surface facing outward. The Li ions in the $WO_3$ film were first driven by the rightward Lorentz force, moved horizontally to the right and accumulated at the right end of the $WO_3$ film. At this moment, the right end of the $WO_3$ film was dark blue, while the rest of the film was transparent. According to the electrochromic mechanism[45], electrons in the conduction band of the metal oxide accordingly moved to the

right end toward the Li ions. In the second cycle, the current was reversed, and the Li ions moved horizontally to the left and accumulated at the left end of the $WO_3$ film. Both cycles correspond to R-L moved state of the $WO_3$ film. In P3, the horizontal voltage was removed. The accumulated Li ions spontaneously diffused into the body of the $WO_3$ film and after 20 min, they uniformly distributed again. At this moment, the entire film was uniformly blue, corresponding to the 20 min self-diffusion state. The morphological evolution of the $WO_3$ film corresponding to the four states confirmed the outlined transport processes of Li ions in the $WO_3$ film (Supplementary Fig. 2). Furthermore, in situ characterization of both the transmittance of the three selected areas of the film (Supplementary Fig. 3) and the step profile of the left-most end of the film (Supplementary Fig. 4) during the entire process clearly illustrated the transport process based on volume expansion and transmittance variation.

Strikingly, we noted that the Li ions moved in the $WO_3$ film along different paths in the first two cycles (Fig. 3a). At the beginning of the first cycle (Fig. 3a), the $WO_3$ film was uniformly blue. When the Li ions were driven and moved to the left in the $WO_3$ film, they appeared to be drawn as if by a pump and moved straight to the left end of the $WO_3$ film. In the second cycle (Fig. 3a), the current flowed to the right and the Li ions were driven from the left-concentrated state. At the beginning, the left-most end of the $WO_3$ film first became transparent, changing from the dark blue state, while the right-most end changed to dark blue from the transparent state. However, the middle area of the film was merely light blue. Next, Li ions seemed to be drawn as if by a pump and moved straight to the right end of the $WO_3$ film just as in the first cycle. In the third cycle (Fig. 3a, 3rd cycle), the movement path of the Li ions was the same as that in the second cycle. In both the second and third cycles, the Li ions were driven from a one end-concentrated state rather than from the uniformly distributed state as in the first cycle. Thus, the transport process in all the following cycles should be the same as that in the second cycle, which has been verified.

Regarding the current-driving model at the microscale (Fig. 3b), in the first cycle, the Li ions at the interface that just escaped into the electrolyte from the $WO_3$ film were first triggered by the Lorentz force and moved one step to the left. Their former places

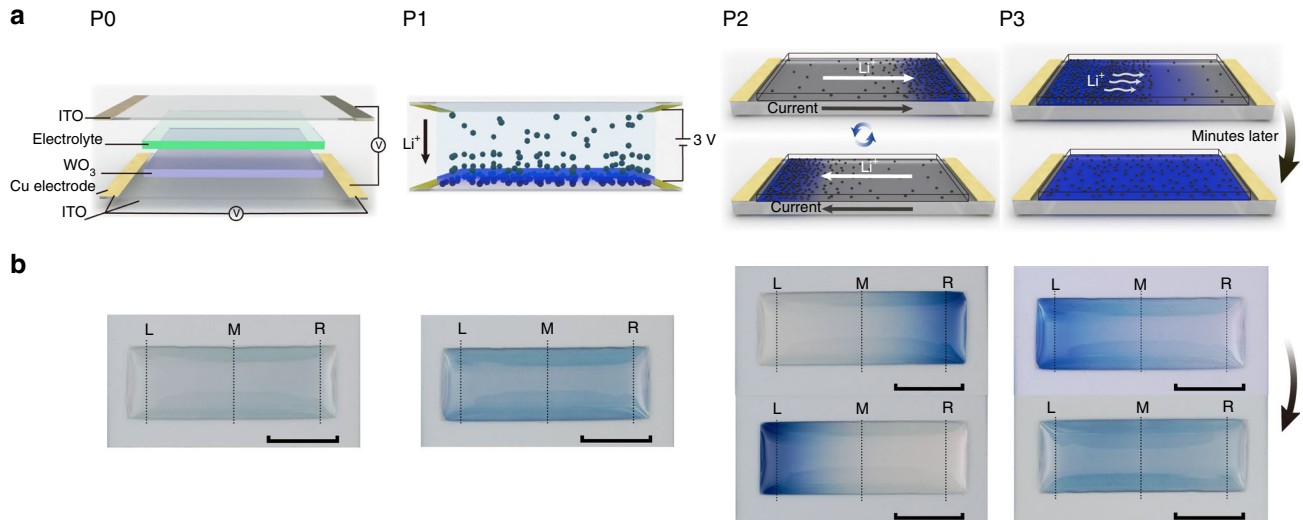

**Fig. 2** Experimental process and phenomena of controllable 2D movement of Li ions in $WO_3$. **a** Illustration of the entire process including P0, P1, P2 and P3, corresponding to Original state, Z-moved state, R-L moved state and the 20 min self-diffusion state of the $WO_3$ film, respectively. **b** Optical photos of the $WO_3$ film corresponding to the four processes. Scale bars, 1 cm. L represents left, M represents middle and R represents right. L, M, and R indicate the monitored areas of the film transmittance, as shown in Supplementary Fig. 3

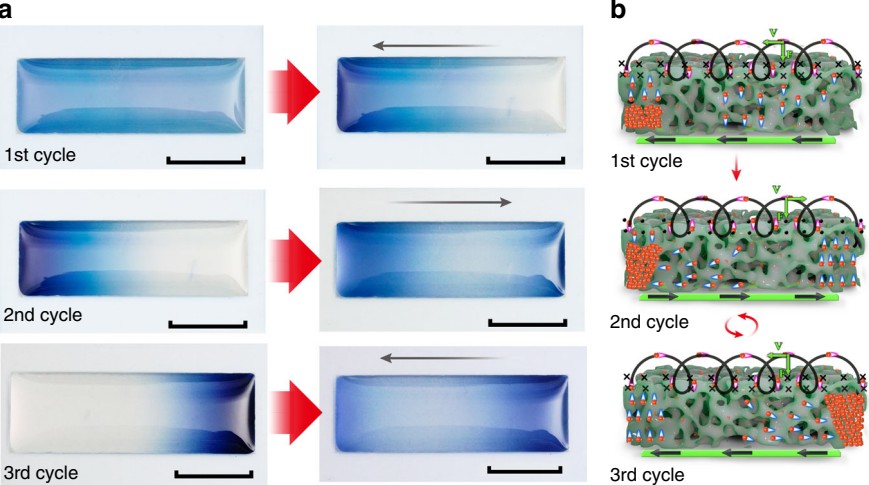

**Fig. 3** Directivity of the 2D movement of Li ions in the WO₃ film during the first two cycles and its consistency with the 'current-driving model'. **a** Optical photos taken at the beginning of each cycle. Scale bars, 1 cm. Black arrows denote the current direction. **b** Interpretation of the directivity with the 'current-driving model'. The directions of the current-induced magnetic fields at the solid–liquid interfaces are denoted with conventional symbols in black: "(X)" for the into-paper-plane and "(●)" for the out-of-paper-plane. The driving current was 0.2 A

were taken by the closely-following Li ions coming from the right end. However, for the right-most end, no Li ions followed from the right, only from the body of the WO₃ film. Therefore, at the right-most end of the WO₃ film, the Li ions appeared to be constantly drawn as if by a pump and directly transported to the left end of the WO₃ film through the interface "passage", while the rest of the WO₃ film seemed to be uninfluenced. When Li ions in the rightmost end were transported, Li ions in the left area next to the rightmost end would then undergo this process. When these Li ions were pumped, Li ions in the next area to the left would then undergo this process. This process would proceed, appearing as if the Li ions in the WO₃ film were directly moving to the left, similar to a curtain being pulled open.

In the second cycle (Fig. 3b 2nd cycle), at the moment the current was reversed, the Li ions accumulated in the left end of the WO₃ film moved along three paths: (1) the Li ions at the interface were driven by the Lorentz force and directly transported to the right-most end of the film through the interface "passage"; (2) some of the Li ions accumulated in the body, diffused upward and joined the moving flow; and (3) the rest of the Li ions in the body, spontaneously diffused rightward to the middle area of the film due to the very large concentration difference and removal of the Lorentz force pointing to the left. Therefore, the Li ions in the left-most end were first transported. Some reached the right-most end of the film, while other ions that diffused to the middle area of the film were waiting to join the moving flow. At this moment, the transport process proceeded in the same way as in the first cycle. The directivity of the transport process in the first two cycles is consistent with the 'current-driving model', which demonstrates that the controllable 2D movement of Li ions in WO₃ occurs not through the body of the metal oxide but through the interface "passage".

**Extraction of the key elements**. Further explorations were performed to determine the key elements in the 'current-driving model' (Fig. 4, Supplementary movie 1). In Model I, first, the WO₃ film was vertically injected with Li ions and turned blue. Electrodes were placed at each end of the blue WO₃ film. An electrolyte was coated onto the middle part of the film surface. Then, a horizontal voltage was applied onto the electrodes, namely, on the WO₃ film. At this moment, no current flowed through the bottom ITO layer, and due to the ultrahigh resistance of the blue WO₃ film (~0.4 MΩ,

see Methods and Supplementary Fig. 5), no current flowed through the WO₃ film either. Hence, a magnetic field was excluded and only an electric field existed. The 2D movement of Li ions in the WO₃ film was not observed. In Model II, two electrodes were placed at each end of the ITO conducting layer. However, no electrolyte was coated on the surface of the WO₃ film. A horizontal voltage was applied on the electrodes, and a constant current flowed through the bottom ITO conducting layer. A magnetic field was generated, which at the solid–liquid interface, was perpendicular to the paper surface facing inward. However, this model also failed. Next, an electrolyte was added to the film surface (Model III), and the 2D movement of Li ions in WO₃ was observed, which demonstrates that the dynamic exchange of Li ions at the interface is a key element. In Model IV, another piece of ITO glass was laid on top of the electrolyte and kept in close contact with the electrolyte. A constant current flowed leftward through the upper ITO layer. Compared with Model III, although the current and the electric field were in the same direction, at the solid–liquid interface, the generated magnetic field was reversed. The Li ions in the WO₃ film moved to the right and accumulated at the right end in Model IV, completely contrary to Model III, suggesting that the 2D movement of Li ions in the WO₃ film was driven by the current magnetic field which is another key element. Notably, an isolated magnetic field does not work, because work has to be done on the system in the 2D transport process which is the third key element.

By manipulating the key elements in the 'current-driving model', the migration distance, the migration rate, the migration region, the migration path and the redistribution of the Li ions in metal oxides are all controllable (Fig. 5). The larger the current strength is, the longer the migration distance and the faster the migration rate (Fig. 5a and b). Li ions moved in zigzags following a "2" pattern when the bottom conducting ITO layer was designed with a "2" pattern on the glass substrate and the WO₃ film was deposited over the entire glass substrate (Fig. 5c and another pattern in Supplementary Fig. 6). The migration area could be selected through control of the electrolyte distribution, because Li ions only moved in regions where the electrolyte was coated (Fig. 5d and Supplementary movie 2). More strikingly, the final distribution of the Li ions after the 2D movement in WO₃ films can be maintained as long as the electrolyte is removed. We painted a smiley face and a silly face in the blue background by manipulating the redistribution of Li ions in WO₃ films (Fig. 5e).

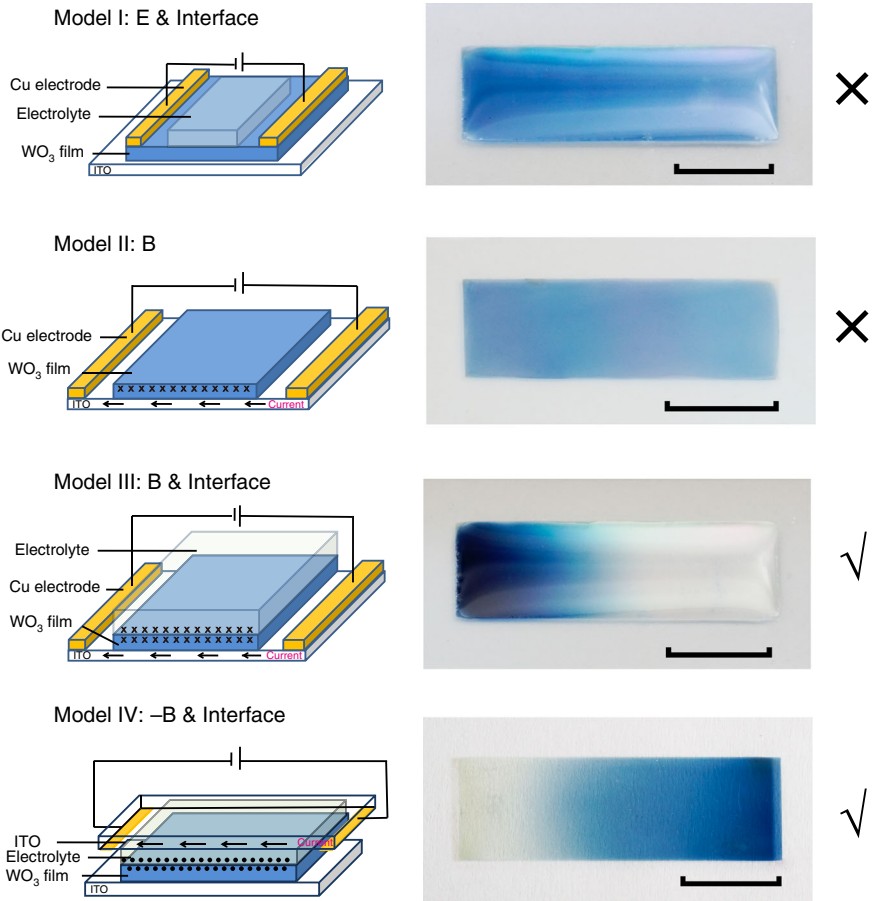

**Fig. 4** Extraction of the key elements in the 'current-driving model'. "WO₃ film" in the schematic diagrams denotes the Z-moved state of WO₃ films. Directions of the current-induced magnetic fields at the solid–liquid interfaces are denoted with conventional symbols in black: "(X)" for the into-paper-plane and "(●)" for the out-of-paper-plane. Scale bars, 1 cm

## Discussion

In summary, we achieved controllable 2D movement and redistribution of Li ions in metal oxides by the 'current-driving model', and control of the transport process and the redistribution was clearly delineated through the color change due to Li ions moving in WO₃ films. The 'current-driving model' has three prerequisites: a solid–liquid interface with dynamic exchange of Li ions, a magnetic field and work being done on the system. Surprisingly, this model also works for other cations, and to date, four other cations have been studied; namely, $H^+$, $Na^+$, $Zn^{2+}$ and $Ca^{2+}$ (see Supplementary Table 2). This facile but effective approach can be extended to deliberately fabricate very large concentration differences to simulate and in situ monitor the transport process of Li ions in metal oxides with uneven thicknesses (Supplementary Fig. 7 and Supplementary movie 3). Additionally, the breakthrough of the controllability of the transport process and the redistribution of Li ions in metal oxides will bring about a great structural revolution from the traditional face-to-face structure to new shoulder-by-shoulder structures, such as planar electrochromic devices (see Supplementary Fig. 8, in this brand-new planar electrochromic smart window, not only the counter electrode layer is omitted but also the window area of the electrochromic film can be customized through 'the current-driving model' like a curtain being pulled open).

## Methods

**Sample fabrication and the current-driving operation.** Amorphous WO₃ films were fabricated by thermal evaporation of pure WO₃ powder (1 g) sprinkled on a

W boat, which was heated for 20 min under the conditions of high vacuum ($2.0$–$4.0 \times 10^{-3}$ Pa) and an electric current of 130 A. No substrate heating was used during the film deposition. The film thickness was different depending on the relative location between the source and the substrate. However, in this study, the film thickness (in the range of 200 nm–1 μm) had no influence on the achievement of the 2D movement of Li ions in metal oxide films. Therefore, the WO₃ films used in this study were not of the same thickness. A film was randomly selected to carry out the film characterization (scanning electron microscopy (SEM) morphology and X-ray diffraction (XRD) spectra in Supplementary Fig. 1). The substrates used in this work were clean glass plates with transparent and electrically conducting layers of $In_2O_3$: Sn (known as ITO) with a sheet resistance of ~8 Ω square$^{-1}$ at a thickness of ~200 nm (Supplementary Fig. 1). Between the glass substrate and the ITO conducting layer was a $SiO_2$ barrier film, whose thickness was approximately 30 nm, as given in the product report. The resistance of the ITO glass substrate (20 mm in width ×50 mm in length) used in this work was approximately 20 Ω as measured by a digital multimeter, and the current flowing through the ITO layer can be adjusted by tuning the applied horizontal voltage. Meanwhile, a 1 mol L$^{-1}$ Li ion electrolyte was prepared by dissolving $LiClO_4$ in a polycarbonate (PC) solution. Then, the prepared ITO/WO₃ samples were coated with the electrolyte with two uncoated ends for the Cu electrodes. Then, another piece of ITO glass was placed onto the electrolyte as the top electrode for vertical injection of Li ions into the film. The device was then ready for the current-driving operation.

First, a 3 V voltage was vertically applied from the upper ITO layer (positive) to the bottom ITO layer (negative), and the WO₃ film was vertically injected with Li ions and turned blue. Second, a voltage was horizontally applied to the two Cu electrodes, and a constant current flowed through the bottom ITO layer. At this moment, a magnetic field was generated, and the injected Li ions moved horizontally in the WO₃ film. In the end, the horizontal voltage was removed. The Li ions spontaneously diffused in the WO₃ film and returned to the uniformly distributed state after approximately 20 min.

**Characterization of the 2D movement and the redistribution of Li ions in metal oxides.** In this study, the entire process of the 2D movement of Li ions in

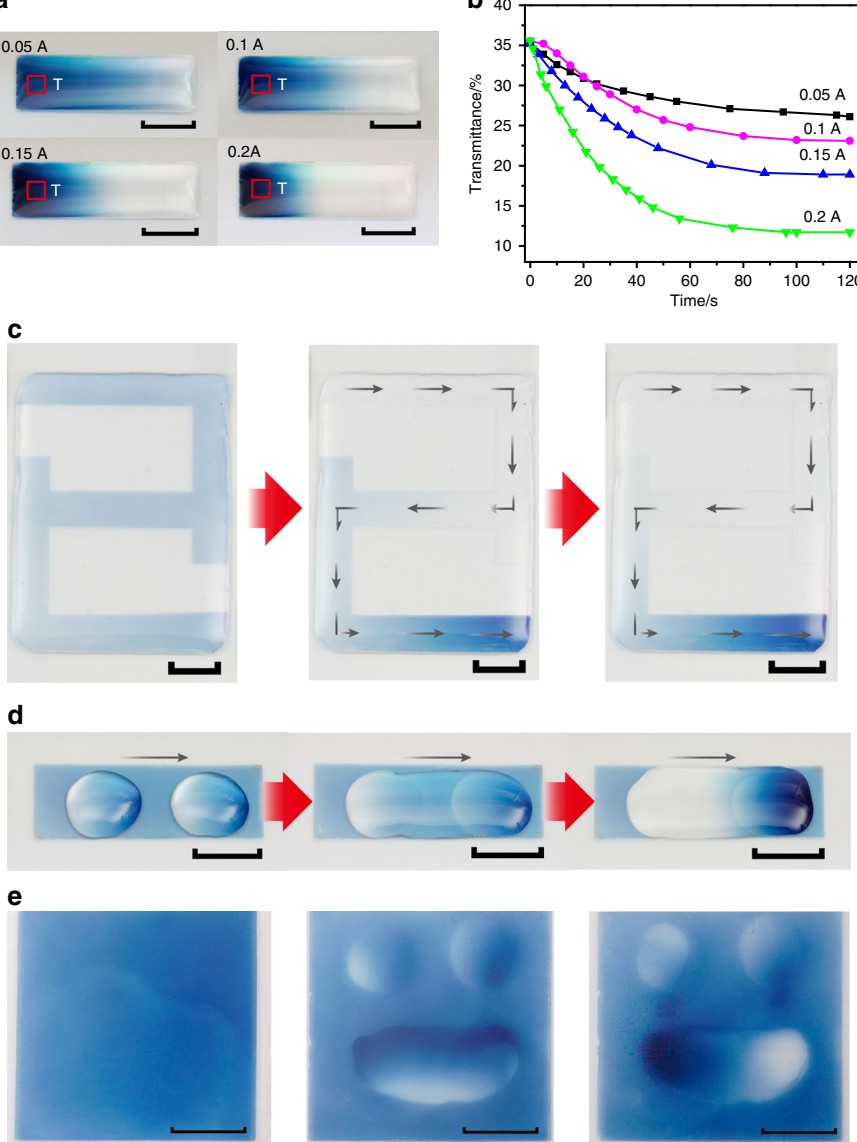

**Fig. 5** Controllable 2D movement and redistribution of Li ions in the $WO_3$ film. Scale bars, 1 cm. **a** Control of the migration distance by the strength of the driving current. Square T denotes the testing area of the film transmittance shown in (**b**); **b** Control of the migration rate by the strength of the driving current; **c** Control of the movement path of Li ions by the current flowing path. The driving current was 0.02 A; **d** Control of the migration area of Li ions by the electrolyte distribution. The driving current was 0.2 A; **e** Images obtained by redistributing Li ions in $WO_3$

metal oxides was referred to as $P_0$, $P_1$, $P_2$ and $P_3$, corresponding to the four states of the metal oxide film, namely Original state (the original film), Z-Moved state (after Li ions were vertically injected), R-L Moved state (after the injected Li ions were driven and horizontally moved) and the 20 min self-diffusion state (the moved Li ions returned to the uniformly distributed state after 20 min self-diffusion), respectively. Optical photos, film transmittance, step profile, SEM morphology, and structural observations were performed on the metal oxide films in the four states to monitor the 2D movement of Li ions in metal oxides.

All optical photos were taken with a digital SRL camera (Model: Sony A6500) and the same film was used for the entire process. The transmittance spectra of the $WO_3$ film on the ITO glass substrate were in situ recorded using a transmission meter (Model: LS162, Linshang Technology Co., Ltd. Shenzhen, China). Tapping-mode atomic force microscopy (AFM) images were taken on the edge of the step between the $WO_3$ film and the ITO glass substrate using a Bruker Dimension Fastscan, and the step profile was in situ recorded during the entire process. The scanning range was 10 μm across the step. The film thickness was calculated from the step profile. Scanning electron microscopy images were taken with a Sigma 500 instrument (Zeiss) at 10.0 kV. For the R-L moved state, the film was dried after the electrolyte was removed when the 2D movement of Li ions was finished. At this time, the distribution of Li ions in metal oxides was maintained and ready for SEM characterization. X-ray diffraction spectroscopy was performed by using an X-ray source of Cu Ka radiation (XRD, X'pert Pro).

## Data availability
The data that support the findings in this study are available upon reasonable request to the corresponding author.

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

## Acknowledgements

We thank Xihong Lu and Weiguang Xie for providing the TiO$_2$ and MoO$_3$ film samples. We also appreciate the help from Jun Zhou and Jiong Zhang in improving and submitting the manuscript. Most of the work is carried out in the School of Applied Physics and Materials and Research Center of Flexible Sensing Materials and Devices in Wuyi University. It is mainly supported by Guangdong Natural Science foundation for Distinguished Young Scholar (2015A030306031), Natural Science Foundation of Guangdong Province (2018A030313561) and Innovation and Strong School Engineering Fund of Guangdong Province (2016KQNCX169 and 2017KTSCX186). The in situ AFM characterization was done in Instrumental Analysis & Research Center of Sun Yat-sen University and was supported in part by Science Foundation for Young Teachers of Wuyi University (No. 2018td04).

## Author contributions

J.L. conceived and supervised the project. X.T. designed the experiments, and jointly performed all data analysis. G.C., Z.M., D.M., and S.W. performed the experiments, including materials synthesis and phenomena collection. L.G. and J.W. conducted the AFM and SEM & XRD analysis respectively. L.Z., J.H., and T.H. contributed in the current-driving model discussion. All authors have given approval to the final version of the manuscript.

## Additional information

**Competing interests:** The authors declare no competing interests.

