## [Peer Review File · Nature Communications]

Reviewers' comments:

Reviewer #1 (Remarks to the Author):

In the submitted manuscript, the authors demonstrated an achievement of controllable 2D movement and redistribution of Li ions by magnetic field. Control of the Li ions is important issues for a lot of electrochemistry field. Authors suggested new method to control the Li ions movement. It is well-written and contains many new findings, which would benefit the readers in this field. So I think it is worth to publish nature communications. However, I would like the authors to check the followings.

- 1) Authors showed control of Li ions by magnetic field by a constant current. If the sample is non-conducting material, is your method not applicable? As you know, most of the electrode material of batteries is almost non-conducting materials. Can your method be applied to battery systems?
- 2) If your method can be applied to the battery systems, could you explain in the manuscript how can control the Li ion movement using magnetic field? You can understand that your method can be applied to electronic systems, but it is not clear whether your method is also useful for battery systems.
- 3) Authors explained the 'current-driving model' in Figure 1. This model was made by you? If not, you would better add appropriate references.

Reviewer #2 (Remarks to the Author):

This article reports a technique for the visualization of lithium diffusion in liquid electrolytes by exploiting the electrochromatic property of WO₃ tungsten oxide. While the reported experiments are interesting and the supplemental movies are clear, the manuscript leaves much to desire and lacks a proper structure. There is no introduction with motivation of the reported research (only general statements are made), a discussion of related work is missing, and so is an assessment of the relevance of the reported findings. The methods section referred to several times in the text also seems to be absent. Additionally, the manuscript suffers from language issues, which makes parts of it hard to understand.

The concept of electrochromic windows that are switched through lithium diffusion in contacted electrolytes does not seem to be novel. For example, in 1991 Baudry et al. reported a device with WO₃ deposited on ITO and a polymer lithium electrolyte (J. Electrochem. Soc. 138, 1991, 460-465).

Considering the quality of the manuscript and the uncertain novelty I can currently not recommend publication of this article.

List of Changes and Responses to the Referees' comments

We are sincerely grateful to the referees for their careful review of our manuscript (Title: *Controllable two-dimensional movement and redistribution of Li ions in metal oxides*. No. *NCOMMS-18-37644-T*). Their constructive comments are helpful and inspiring for us to reword some of the statements in the manuscript to make it precise, clear and easy to read. The point-by-point responses to their questions/concerns are listed as follows.

Reviewer #1

In the submitted manuscript, the authors demonstrated an achievement of controllable 2D movement and redistribution of Li ions by magnetic field. Control of the Li ions is important issues for a lot of electrochemistry field. Authors suggested new method to control the Li ions movement. It is well-written and contains many new findings, which would benefit the readers in this field. So I think it is worth to publish nature communications. However, I would like the authors to check the followings.

Response: Thanks very much for your careful review and positive comments.

Comment (1): Authors showed control of Li ions by magnetic field by a constant current. If the sample is non-conducting material, is your method not applicable? As you know, most of the electrode material of batteries is almost non-conducting materials. Can your method be applied to battery systems?

Response: We truly understand the referee's concern. In this work, planar movement and redistribution of Li ions in metal oxides was achieved manipulation by a magnetic

field that was generated by a constant current. The current was flowing through the bottom conducting layer beneath the metal oxide layer rather than the metal oxide layer (see Fig.R1). To date, our model has been verified to be applicable to four kinds of metal oxides, including WO_3 , MoO_3 , TiO_2 and $\text{Nb}_2\text{O}_5 @ \text{graphite}$ (see Supplementary Information Table S1). Among them, WO_3 , MoO_3 and TiO_2 are all non-conducting materials. So, our model is applicable to non-conducting material.

What's more, achievement of applying the model on the $\text{Nb}_2\text{O}_5 @ \text{graphite}$ nanocomposite on copper foil that was used as potential anode material in Li-ion batteries, demonstrated our model is applicable to the anode part of the battery system. Further and systematic work on a whole battery system is still being investigated. Meanwhile, a brand-new shoulder-by-shoulder battery structure based on this model is also under investigation, in which, completely different from the traditional face-to-face structure, anode and cathode materials are fabricated shoulder by shoulder on the same collector substrate and Li ions are rocked between the two electrodes by an extra magnetic field.

Fig. R1 Schematic illustration of the experimental set-up in our manuscript.

Comment (2): If your method can be applied to the battery systems, could you explain in the manuscript how can control the Li ion movement using magnetic field?

You can understand that your method can be applied to electronic systems, but it is not clear whether your method is also useful for battery systems.

Response: This is a very constructive question and it is exactly the motivation of this work. Notably, the experimental set-up of this ‘current-driving model’ (see Fig.R1) is exactly the layout of the collector and the electrodes in a battery system. So the model was tried on the potential anode material Nb₂O₅ @ graphite nanocomposite on copper foil, using LiClO₄ in PC solution as the electrolyte. First, Li ions were intercalated into the nanocomposites by a vertical voltage applied between the upper conducting layer and the bottom Cu foil. Then, a horizontal voltage was applied onto each end of the Cu foil and a constant current was flowing through it. A magnetic field was generated by the flowing current. It turned out the intercalated Li ions in the Nb₂O₅ @ graphite nanocomposite were moved and redistributed (see Supplementary Information Table S1), suggesting the model is applicable to the anode part of the battery system. Thus based on the model, the movement and the redistribution of the intercalated Li ions in anode metal oxides could be manipulated by the current strength, the current’s flowing pattern, or the electrolyte distribution (see Fig.5 in the manuscript). Further exploration of applying the model on more electrode materials and on a whole battery system needs the concerted efforts of us all and more time.

Comment (3): Authors explained the ‘current-driving model’ in Figure 1. This model was made by you? If not, you would better add appropriate references.

Response: The ‘current-driving model’ was indeed proposed by us. We were longing for manipulating the intercalated Li ions in electrodes, including their movement and

the redistribution. We have been making continuous efforts in this. Enlightened by the ubiquitous phenomena “self-discharge” in batteries and “self-bleaching” in electrochromic devices, the solid-liquid interface was deeply analyzed. And to control the Li ions by using a magnetic field came to our mind. Considering the second law of thermodynamics, work must be done on the system and a magnetic field generated by a constant current is preferred, referred to as the ‘current-driving model’.

Reviewer #2

Comment (1): This article reports a technique for the visualization of lithium diffusion in liquid electrolytes by exploiting the electrochromatic property of WO₃ tungsten oxide.

Response: Thanks for your patient work. Indeed, in this work visualization of the movement and redistribution of Li ions was reported. And this movement and redistribution of Li ions both happened in metal oxides not in liquid electrolyte. More importantly, in our work this movement and redistribution of Li ions in metal oxides is not a simple diffusion process, but for the first time it was achieved to be manipulated by an external force. Therefore, our work aimed to report the achievement of the controllable planar movement and the redistribution of Li ions in metal oxides after they have been intercalated, through the ‘current-driving model’. This model has three prerequisites: a solid-liquid interface with dynamic exchange of Li ions, a magnetic field and work being done on the system. To date, this model has been verified to be applicable in four types of metal oxides, including WO₃ films,

MoO₃ films, TiO₂ nanorod films and Nb₂O₅ @ graphite nanocomposites (see Supplementary Information Table S1). Owing to the visualization of Li ions moving in WO₃ films (namely the electrochromic property of WO₃), WO₃ film was only taken as an example and delineated in detail in the manuscript.

Comment (2): While the reported experiments are interesting and the supplemental movies are clear, the manuscript leaves much to desire and lacks a proper structure.

Response: Thank you for your careful review and positive comments. It should be noted that this manuscript was transferred from the journal of Nature Materials (NM) and thus it was organized according to the format demands of the NM. That's why it looked like lacking a proper structure. Now, the structure of the manuscript has been readjusted according to the format requirements of Nature Communications.

Comment (3): There is no introduction with motivation of the reported research (only general statements are made), a discussion of related work is missing, and so is an assessment of the relevance of the reported findings.

Response: In this work, we achieved to manipulate the planar movement and the redistribution of intercalated Li ions in metal oxides through an external force. The discussion of related work was mainly arranged at the second paragraph of the manuscript (ranging from Line 12, Page 2 to Line 3, Page 3). With the formation of the intercalation theory and the appearance of the secondary lithium batteries, intensive research has been directed toward monitoring the diffusion movement and mapping the spatial distribution of Li ions in electrode materials to provide fundamental understandings about the transport and accumulation process of the

intercalated Li ions (see references 25-29). What's more, the latest progresses were reworded and highlighted according to this comment, which has been marked in red in the manuscript (see references 30-31).

Comment (4): The methods section referred to several times in the text also seems to be absent.

Response: Thank you for your careful review and helpful comments. Because this manuscript was first submitted to the journal of NM, the Methods section had been put in the supplementary formation according to its format requirement. Now, the Methods section has been put in the manuscript according to the format requirements of Nature Communications.

Comment (5): Additionally, the manuscript suffers from language issues, which makes parts of it hard to understand.

Response: Thank you for your sincere and helpful opinion. We have sent our manuscript for professional editing service using Springer Nature Author Services. Now some of the statements in the manuscript have been reworded accordingly. We hope now it is precise, clear and easy to read. All revisions were marked in red in the manuscript. The service certification was pasted in the end as Attachment one.

Comment (6): The concept of electrochromic windows that are switched through lithium diffusion in contacted electrolytes does not seem to be novel. For example, in 1991 Baudry et al. reported a device with WO_3 deposited on ITO and a polymer lithium electrolyte (J. Electrochem. Soc. 138, 1991, 460-465).

Response: In this work, we aimed to report the manipulation of the planar movement

and the redistribution of Li ions in metal oxides. WO₃ film was only taken as an example and delineated in detail in the manuscript, owing to the visualization of Li ions moving in WO₃ films that is also referred as the electrochromic property of WO₃ films. But both the electrochromic property of WO₃ films and the PC LiClO₄ electrolyte were only used as a tool to interpret the detailed experimental set-up and vividly display the moving process and accumulated state of Li ions in metal oxides.

Additionally, based on this work, a new concept of planar electrochromic smart window was proposed in the end of the manuscript and shown in Supplementary information Fig.S8. As we know, in traditional electrochromic smart windows, Li ions in the electrolyte are vertically driven by an electric field that is applied between the upper and the bottom transparent conducting layers, and intercalated into the electrochromic films. Then the whole electrochromic films get colored. However, in this new planar electrochromic smart window, not only the counter electrode layer is omitted, but also the window area of the electrochromic film can be customizable through the current-driving model like a curtain being pulled open.

Comment (7): Considering the quality of the manuscript and the uncertain novelty I can currently not recommend publication of this article.

Response: In this work, breakthrough of the controllability of the transport process and the redistribution of Li ions in metal oxides was reported for the first time. We believe this breakthrough would make great potential influences in the field of electrochemistry and many more waits to be determined. We have tried our best to improve this manuscript according to your constructive comments. Sincerely, we hope

now it is satisfying and could be published on this influential journal of Nature Communications. Thus, with the the concerted efforts of us all, it can play a role in enhancing the development of Li-ion batteries. Thank you very much!

Attachment One:

SPRINGER NATURE | Author Services

Nature Research Editing Service Certification

This is to certify that the manuscript titled Controllable two-dimensional movement and the redistribution of Li ions in metal oxides was edited for English language usage, grammar, spelling and punctuation by one or more native English-speaking editors at Nature Research Editing Service. The editors focused on correcting improper language and rephrasing awkward sentences, using their scientific training to point out passages that were confusing or vague. Every effort has been made to ensure that neither the research content nor the authors' intentions were altered in any way during the editing process.

Documents receiving this certification should be English-ready for publication; however, please note that the author has the ability to accept or reject our suggestions and changes. To verify the final edited version, please visit our verification page. If you have any questions or concerns over this edited document, please contact Nature Research Editing Service at support@as.springernature.com.

Manuscript title: Controllable two-dimensional movement and the redistribution of Li ions in metal oxides

Authors: Xiufeng Tang, Guoxin Chen, Zhaopeng Mo, Dingbang Ma, Siyuan Wang, Jinxiu Wen, Li Gong, Lite Zhao, Jingcheng Huang, Tengcheng Huang, Jianyi Luo

Key: 0A46-EC12-26F1-1964-13BP

This certificate may be verified at secure.authorservices.springernature.com/certificate/verify.

Nature Research Editing Service is a service from Springer Nature, one of the world's leading research, educational and professional publishers. We have been a reliable provider of high-quality editing since 2008.

Nature Research Editing Service comprises a network of more than 900 language editors with a range of academic backgrounds. All our language editors are native English speakers and must meet strict selection criteria. We require that each editor has completed or is completing a Masters, Ph.D. or M.D. qualification, is affiliated with a top US university or research institute, and has undergone substantial editing training. To ensure we can meet the needs of researchers in a broad range of fields, we continually recruit editors to represent growing and new disciplines.

Uploaded manuscripts are reviewed by an editor with a relevant academic background. Our senior editors also quality-assess each edited manuscript before it is returned to the author to ensure that our high standards are maintained.

Reviewers' comments:

Reviewer #1 (Remarks to the Author):

It could be published as it is.

Reviewer #2 (Remarks to the Author):

In the revision and the rebuttal letter, the authors have addressed my concerns regarding the novelty of their findings, and the copyediting has significantly improved the clarity of the manuscript. It seems that in-plane control of lithium diffusion using magnetic fields indeed has not been previously reported. Once the authors have addressed the issues below, I am inclined to recommend the manuscript for publication.

1. Clarity of the schematics

The details of the schematics in figure 1 and 3 are too small to be useful. While I think that all information is depicted, the arrows are just too small on a printout. I suggest to more clearly indicate (i) the direction of the assumed ejection velocity at the solid liquid interface using a sufficiently large arrow, (ii) the direction of the induced magnetic field with conventional symbols, i.e., "(X)" for into-paper-plane and "(·)" for out-of-plane (this would be especially useful in figure 4), and (iii) the direction of the resulting Lorentz force. The arrow that indicates the direction of the current should also be colored in a clearly distinguishable way (for example in solid black). It would help to reduce the color of the metal oxide in these schematics, so that other features become more clearly visible.

2. Comparison with conventional electrochromic windows

I understand that WO₃ is mainly used as a tool for visualizing the motion of Li in this paper. However, especially for readers from the battery community more background about electrochromic materials is needed. The difference between the conventional two-electrode approach and the here introduced single-electrode approach is discussed in the rebuttal letter and should be included in the main manuscript along with references.

List of Changes and Responses to the Referees' comments

We are sincerely grateful to the referees for their careful review of our manuscript (Title: *Controllable two-dimensional movement and redistribution of Li ions in metal oxides*. No. NCOMMS-18-37644A). The comments are very helpful in improving the manuscript. The point-by-point responses to the concerns are listed as follows.

Reviewer #1

It could be published as it is.

Response: Thanks very much for your careful and patient work.

Reviewer #2

In the revision and the rebuttal letter, the authors have addressed my concerns regarding the novelty of their findings, and the copyediting has significantly improved the clarity of the manuscript. It seems that in-plane control of lithium diffusion using magnetic fields indeed has not been previously reported. Once the authors have addressed the issues below, I am inclined to recommend the manuscript for publication.

Response: Thanks very much for your patient work and helpful and positive comments.

Comment (1): Clarity of the schematics.

The details of the schematics in figure 1 and 3 are too small to be useful. While I think that all information is depicted, the arrows are just too small on a printout. I suggest to more clearly indicate (i) the direction of the assumed ejection velocity at the solid liquid interface using a sufficiently large arrow, (ii) the direction of the induced magnetic field with conventional symbols, i.e., "(X)" for into-paper-plane and "(•)" for out-of-plane (this would be especially useful in figure 4), (iii) the direction of the resulting Lorentz force. The arrow that indicates the direction of the current should also be colored in a clearly distinguishable way (for example in solid black). It

would help to reduce the color of the metal oxide in these schematics, so that other features become more clearly visible.

Response: Thanks very much for this careful and constructive comment. We greatly revised Fig.1, Fig.3 and Fig.4 in the manuscript accordingly. Now, details in the schematic shown in Fig.1 and Fig.3 are much clearer,

- (i) the direction of the assumed ejection velocity at the solid liquid interface has been denoted using large and obvious symbols;
- (ii) the direction of the induced magnetic field at the solid-liquid interface has been illustrated with conventional symbols in black "X";
- (iii) the direction of the resulting Lorentz force has been shown in the schematic as "F";
- (iv) the arrow that indicates the direction of the current has been colored in solid black;
- (v) the color of the metal oxide in the schematic has been reduced to be light green.

What's more, the directions of the induced magnetic field at the solid-liquid interface with conventional symbols in black ("X" for into-paper-plane and "(•)" for out-of-plane) were added in Fig.4.

Comment (2): Comparison with conventional electrochromic windows

I understand that WO_3 is mainly used as a tool for visualizing the motion of Li in this paper. However, especially for readers from the battery community more background about electrochromic materials is needed. The difference between the conventional two-electrode approach and the here introduced single-electrode approach is discussed in the rebuttal letter and should be included in the main manuscript along with references.

Response: Thank you for your careful review and such helpful comment. We have revised the manuscript accordingly. Statements about the electrochromism and the difference between the conventional two-electrode approach and this work have been added in the main manuscript (Page 6, Line 5-Line 20 and Page 15, Line 15-Line 18). Along with the statements, one more related reference was added as Reference 45.